# Body Image and Disturbed Eating Attitudes and Behaviors in Sport-Involved Adolescents: The Role of Gender and Sport Characteristics

**DOI:** 10.3390/nu11123061

**Published:** 2019-12-14

**Authors:** Rasa Jankauskiene, Miglė Baceviciene

**Affiliations:** 1Institute of Sport Science and Innovation, Lithuanian Sports University, Sporto 6, 44221 Kaunas, Lithuania; rasa.jankauskiene@lsu.lt; 2Department of Physical and Social Education, Lithuanian Sports University, Sporto 6, 44221 Kaunas, Lithuania

**Keywords:** adolescence, body image, disordered eating, competitive sport, physical activity

## Abstract

Disordered eating in adolescents who participate in sports is an issue of great concern. However, very few studies have examined the prevalence of sport-related determinants of disturbed eating attitudes and behaviours (DEABs) in sport-involved adolescents. The present study aims to assess body image and DEABs in the sample of adolescents involved in a sport of different characteristics (competitive sport, leisure sport; weight-sensitive and less-weight-sensitive sport) and to compare the results with those of the controls. A total sample of 732 adolescents (437 (59.4%) were female) participated in the study. The participants ranged in age from 16 to 19 years (mean = 17.2, SD = 0.6). Study participants completed a questionnaire assessing sports characteristics, body image, disordered eating, and health-compromising eating behaviours. Analysis of covariance was used to test the differences of major study variables between controls and weight-sensitive and less-weight-sensitive leisure and competitive sports groups. There was no significant difference observed in DEABs between the sports groups. Competitive athletes reported more satisfaction with body image than controls. The girls participating in weight-sensitive leisure sports reported higher overweight preoccupation and greater DEABs compared to those participating in less-weight-sensitive leisure sports. Boys participating in weight-sensitive leisure sports reported a greater drive for muscularity-related behaviour compared to those participating in less-weight-sensitive leisure sports. There was no observed interaction between gender and weight sensitivity in the competitive sports group. Adolescents involved in a competitive sport demonstrated greater body image satisfaction and did not seem to present a greater risk for DEABs than controls. Special prevention attempts for lowering body image concerns and DEABs should be addressed for adolescents participating in a weight-sensitive leisure sports.

## 1. Introduction

Body image concerns and body dissatisfaction are highly prevalent in adolescents worldwide [1,2,3,4,5]. Studies have demonstrated that adolescents’ body image concerns are associated with disturbed eating attitudes and behaviors (DEABs) that might be the precursor of clinical eating disorders [6,7]. DEABs are abnormal eating-related attitudes (e.g., fear to gain weight, a drive for thinness or a drive for muscularity) and behaviours (e.g., restricted dietary intake, binge eating, self-induced vomiting, laxative abuse, diuretic use, fasting, skipping meals, avoiding a particular group of food, etc.) [8,9]. Those behaviors are damaging and can affect nearly every system of a growing adolescent’s body [6,10], and be predictors for the development of obesity later in life [11,12].

Physical activity and participation in sports are highly promoted for adolescents because of the numerous benefits for health [13]. However, the results of the studies exploring the associations of body image and sport participation in adolescents are not consistent. Some studies have demonstrated that participation in sports and exercise is associated with a greater body image in adolescents [14,15,16,17]. However, other studies have shown that the prevalence of body image concerns is greater in adolescents involved in weight-sensitive sports [18,19]. Weight-sensitive sports are those in which body weight has high impact on the sports performance, i.e. esthetic (i.e., dancing, gymnastics), weight class (i.e., judo, boxing), gravitational technical (i.e., high jump, triple jump), and gravitational endurance (i.e., swimming, long-distance running) sports [20,21,22]. Since studies of body image concerns in adolescents involved in different types and levels of sports are rare and inconclusive, the present study aimed to expand the knowledge of this issue.

Body image concerns, specifically, body dissatisfaction is the precursor of DEAB’s. DEABs have been increasingly studied in samples of adolescents that participate in sports [4,8,23,24,25]. Studies demonstrated that DEABs contribute to poor health and performance in sport, musculoskeletal injury, the development of female and male athlete triad [26,27,28,29], and drop out from sport [30,31]. Typically, DEABs start to develop among adolescents between ages 14 to 17, which is the same time when athletes begin specializing in a particular type of sport [31]. Unfortunately, the results of the studies on the prevalence of DEABs in sport involved adolescents and controls are inconsistent. Some studies showed a higher prevalence of disordered eating among the controls compared to the athletes [4,24,32,33]. Other studies concluded that athletes competing in weight-sensitive sports reported greater rates of eating pathologies [34,35,36], however, other studies in elite adolescent athletes did not confirm these findings [32,37]. 

The methodological differences between studies might explain these inconsistencies in conclusions. Only a minority of studies examined DEABs in large samples of adolescents of both genders and took into account the moderating effect of gender and the association with the sport-related determinants, such as type of sport (weight-sensitive or less-weight-sensitive) and level of competition (professional or elite sport or leisure sport). Only a small proportion of studies compared the results with matched control groups [8,32]. Moreover, there is a lack of studies that analyzed multifactorial risk factor models for DEABs, especially in boys (including body image, drive for muscularity, social physique anxiety, etc.) [21]. Therefore, we aimed to fill that gap. 

To the best of our knowledge, this is one of the first studies exploring body image and DEABs in adolescents involved in different levels of sports in Eastern Europe. Lithuania, as a post-soviet country, has a unique and well-developed non-formal sports school system with a significant number of adolescents participating in Olympic and non-Olympic sports [38]. Traditionally, this system is highly oriented towards professional sports achievements and early specialization which might be a risk factor for DEABs [32]. Unfortunately, there is lack of scientific data on the body image concerns and DEABs in sports participating adolescents of Eastern Europe. Thus, the findings of the present study might be important for the development of DEAB’s prevention programs in Eastern Europe and worldwide. 

The present study aimed to assess body image and DEABs in the sample of adolescents involved in sports with different characteristics and compare the results with those from adolescents that do not participate in sports. We expected that adolescents participating in sports would demonstrate greater body image and lower DEABs compared to sex-matched controls. Secondly, we hypothesized that participants of the competitive sport would demonstrate greater body image and lower DEABs compared to leisure exercisers and controls. Further, we expected that adolescents participating in weight-sensitive sports would demonstrate greater body image concerns and DEABs compared to participants of less-weight- sensitive sports. Finally, we aimed to verify whether gender moderates the associations between body image, DEABs and the aforementioned characteristics of sports practice.

## 2. Methods

### Procedure and Participants

This study included adolescents in the 11th grade from 16 secondary schools in Kaunas, Lithuania. Schools were randomly selected from the institutional registry list of the Ministry of Education and Science. The study was approved by the Kaunas municipality Education Department (No. 35-2-570) and Ethical Board of Institute of Social Sciences of Lithuanian Academy of Physical Education (protocol No. 3). Written informed consent was obtained from all study participants. Questionnaires were filled out during school lessons by a group of trained researchers. The respondents provided their answers by completing questionnaires consisting of a battery of self-report instruments designed to measure study variables. It took approximately 45 minutes to complete each questionnaire. For this study, 856 questionnaires were completed. 63 participants were excluded from the study because their questionnaires were filled in incompletely with more than 50% of missing responses. Further, four questionnaires were additionally removed from this analysis because respondents did not provide information about exact participation in sports. Next, we decided to analyze the questionnaires only of those adolescents witch sports participation lasted longer than half a year and they could be assigned to the maintenance stage according to Prochaska and colleagues’ Transtheoretical Model [39]. Thus, 57 participants were excluded from the study because their reported sports participation was shorter than half a year. 

A total of 732 adolescents’ (437 or 59.4% female) questionnaires were analyzed in the present study. The participants ranged in age from 16 to 19 years (mean = 17.2 years, SD = 0.6). The sample was divided into two groups based on their regular involvement in sports; 459 (62.8% of the sample) adolescents reported participation in a sport and 273 (37.2%) of adolescents were not involved in any organized or non-organized sports activities. Sport involved group was then divided into 2 groups based on adolescents’ reported nature of sport participation: 220 (30.1%) of adolescents reported that they participate in competitive sports (answer “yes” to the question Do you participate actively in competitive sport and take part in sport competitions with professional sport goals?), and 239 (32.7%) of adolescents reported that they participated in leisure recreation without the intention to compete in competitions (answer “no” to the question Do you participate actively in competitive sport and take part in sports competitions with professional sport goals? and answer “yes” for the question Do you exercise in leisure time without the intention to compete in sports competitions?”). Each of the competitive and leisure-time sports groups were divided into 2 groups (weight-sensitive and less-weight-sensitive sports) according to the proposed classification [20,21,22]. The distribution of the competitive and leisure time sport participants in weight-sensitive and less-weight-sensitive sports is presented in Appendix A and Appendix B accordingly. 

## 3. Measures

### 3.1. Demographics

Participants were asked to indicate their age and date of birth.

Body mass index (BMI) was assessed using self-reported height and weight from which BMI was calculated (kg/m^2^). The mean BMI of the students was calculated as the individual’s weight divided by their height squared. As recommended by the International Obesity Task Force (IOTF) cut-offs, the sample was classified into 4 body mass categories according to percentiles: below the 5th percentile was thin, between the 5th and 84th percentiles was normal weight, between the 85th and 94th percentiles was overweight and above the 95th percentile was obese [40]. In further analyses, the overweight and obesity categories were combined. The sample comprised 13.6% of participants who were classified as underweight, 78.7% had normal weights and 7.7% were overweight or obese. Details of anthropometric and sports participation characteristics are presented in Table 1.

### 3.2. Physical Activity

We examined how frequently physical activity (PA) had been performed for 60 minutes or more per day in the last seven days. This item is used in the WHO questionnaires conducted by the international study of Health Behaviour in School-aged Children.

Eating Attitudes Test 26 (EAT-26) [41] was used to measure the participants’ disordered eating and risk for eating disorders. EAT-26 consists of 26 items scored on a Likert scale (0 = never to 6 = always), and the items are summed to obtain an overall score ranging from 0 to 78. Individuals with an overall score of 20 or higher are considered to be at risk of an eating disorder. EAT-26 consists of 3 subscales: bulimia, food preoccupation and oral control. In this study, we used the scale of 26 items with a higher score indicating a greater risk for eating disorders. Internal consistency of the scale was good, Cronbach α = 0.88. Further analyses were performed in 3 groups: scores of ≤ 10, scores of 10–20 and scores of ≥ 20. The internal consistency of the scale was satisfactory with Cronbach α = 0.89.

Health-compromising eating behaviours (HCEBs) included specific behaviours that were not professionally recommended for weight reduction or are physically damaging [9]. Those behaviours were assessed through the question “Have you ever done any of the following things to lose weight or keep from gaining weight in your life?” The following behaviours were included: 1) skip meals, 2) fast, 3) smoke more cigarettes, 4) eat very little food (less than 800 kcal), 5) use diuretics, 6) eat only one product or liquid while dieting, 7) took diet pills to suppress appetite, 8) made yourself vomit, 9) use laxatives. Each method was rated on a Likert scale (from 1 = “not at all true for me” to 5 = “very true for me”). Reporting the use of any of these weight control behaviours was categorized as engaging in health-compromising eating behaviours, the Cronbach alpha = 0.84, test-retest reliability ICC = 0.87.

The Lithuanian version of the short version of the Multidimensional Body and Self Relations Questionnaire-AS (MBSRQ-AS) was used [42]. This instrument evaluates the appearance-related components of the body image construct. It consists of 7 items on the Appearance Evaluation Scale, which measures feelings of physical attractiveness and satisfaction with one’s looks; higher scores indicate higher appearance evaluation. The 12-item Appearance Orientation Scale assesses the extent of investment in one’s appearance, and higher scores indicate higher appearance orientation. The Overweight Preoccupation Scale assesses fat anxiety, weight vigilance, dieting, and eating restraint and consists of 4 items. Higher scores on this scale show a greater preoccupation with being overweight. The Body Areas Satisfaction Scale (9 items) assesses satisfaction or dissatisfaction with specific areas of the body on a 5-point scale (complete satisfaction to complete dissatisfaction). The Self-Classified Weight Scale consists of 2 items and reflects how one perceives and labels one’s weight (ranging from very underweight to very overweight). A higher score on this scale represents greater beliefs that the subject’s body weight is too high. The Lithuanian version of the MBSRQ-AS has been established in the literature [43]. The scale was obtained from the official site with the official permission of the author. Internal consistency of the scales (Appearance Evaluation, Appearance Orientation, Body Areas Satisfaction, Overweight Preoccupation, Self-Classified Weight) was as follows: 0.73; 0.73; 0.87; 0.76 and 0.76, respectively.

We used the Drive for Muscularity Scale (DMS) [44] to examine adolescents’ behaviours that reflect a preoccupation with muscularity. The scale assesses an individual’s perception that he or she is not muscular enough and that bulk should be added to his or her body frame in the form of muscle mass; this belief is irrespective of a person’s percentage of actual muscle mass or body fat. In this study, we used the Muscle Development Behaviors subscale (e.g., “I use protein or energy supplements”). The subscale consists of 7 items rated on a 6-point scale ranging from 1 (never) to 6 (always). Higher scores indicate greater muscle development behaviors. For the present study, the internal consistency of the scale was satisfied (Cronbach α = 0.89).

The Lithuanian version of Rosenberg’s Self-Esteem Scale [45] is the most widely used measure of global self-esteem and has been determined to be valid and reliable among students. The scale consists of 10 items rated on the 4-point Likert scale (from 1 = “not all true for me” to 4 = “very true for me”) with total scores ranging from 10 to 40. Higher scores reflect a greater level of self-esteem. For the present study, the internal consistency of the scale was satisfactory (Cronbach α = 0.76).

### 3.3. Statistical Analysis

Firstly, descriptive statistics and distribution normality testing of continuous variables were performed. Internal consistency of the questionnaires was calculated with Cronbach’s alpha (α) coefficient, external reliability was tested with an intraclass correlation coefficient (ICC). Secondly, preliminary analyses were conducted to examine differences of major study variables across 3 groups of participation in sports (none, leisure and competitive sports). *χ*^2^ was used to test differences of gender, physical activity and exercise frequency in three sports participation groups. Mann Whitney U test was used to compare the duration of sports participation in leisure and competitive groups. And finally, analysis of covariance was conducted to test differences of major study variables in weight-sensitive and less-weight-sensitive leisure and competitive sports groups. All the comparisons were controlled for body mass index and the effects of gender; weight-sensitivity in sports model statistics and interaction effects of gender and weight-sensitivity were calculated. For this study, the significance threshold considered relevant was *p* < 0.05 and 95% confidence intervals (CIs). Statistical analyses were conducted using IBM SPSS Statistics 25 (IBM Corp., Armonk, NY, USA).

## 4. Results

As summarized in Table 1, a significantly (*p* < 0.0001) higher proportion of girls were not participating in any sports as compared to boys, while boys were more frequently involved in competitive sports. There were no differences found in BMI distributions in three participation in sports groups. In addition, in the group of non -participation in sports the proportion of adolescents was lower in the categories of more frequent PA as compared to those who were physically active less than 1 day per week. As opposite, adolescents participating in leisure and competitive sports demonstrated more frequent PA (*p* < 0.0001). Moreover, the majority of the athletes in the competitive sports group trained 5-7 days per week. In the leisure sports group more frequent exercise was less prevalent (*p* < 0.0001). 65.3% from leisure and 60.9% from competitive groups were involved in weight-sensitive sports (*p* = 0.333). Average duration of involvement in sports was significantly longer in the competitive sports group (*p* = 0.002).

Next, we compared the body images of the adolescents and disordered eating and health- compromising eating behaviours (DEABs) by participation in sports groups while controlling for gender and BMI (Table 2). Results showed that sports-involved adolescents demonstrated significantly greater self-esteem, appearance evaluation, appearance orientation, and body areas satisfaction, yet there were no differences observed in overweight preoccupation and self-classified weight. The appearance evaluation score was significantly higher in competitive sports as compared to non-participating in sports (*p* < 0.001) as well as comparing leisure and competitive sports groups (*p* < 0.05). No differences were observed in DEABs between groups based on levels of sports involvement. However, sport-involved adolescents demonstrated a greater drive for muscularity: significant differences were found when comparing non participating in sports with participating in competitive sports (*p* < 0.001) as well as participating in leisure and competitive sport (*p* < 0.05). We then compared adolescents’ body image concerns and disordered eating across weight-sensitive and less-weight-sensitive leisure sports groups and genders (Table 3). A clear gender interaction within the weight-sensitivity group was found. The female subjects participating in weight-sensitive leisure sports reported greater disordered eating and health-compromising eating behaviors compared to females participating in less-weight-sensitive leisure sports. The gender and weight-sensitivity group interaction was observed in the drive for muscularity. Male subjects in weight-sensitive leisure sports reported a greater drive for muscularity than those participating in less-weight-sensitive leisure sports. Gender moderated the relationships between overweight preoccupation and involvement in weight-sensitive leisure sports. Females participating in weight-sensitive leisure sports demonstrated greater overweight preoccupation compared to those participating in less-weight-sensitive leisure sports.

Further, we compared adolescents’ body image concerns and disordered eating across weight-sensitive and less-weight-sensitive competitive sports groups and genders (Table 4). We observed no statistically significant differences between body image, DEABs, muscularity-driven behaviour in adolescents involved in competitive sports who were participating in weight-sensitive and less-weight-sensitive sports groups. In addition, no gender and weight-sensitivity interaction effects were observed, but scores of DEABs, appearance orientation, overweight preoccupation, and self-classified weight were higher in girls, whereas muscularity development behaviors—in boys participating in competitive sports (*p* < 0.05).

## 5. Discussion

The present study aimed to assess body image and DEABs in a sample of late adolescents involved in sports practice of different characteristics and to compare the results with adolescents that did not participate in sports. We expected that adolescents participating in sports would demonstrate greater body image and lower DEABs compared to sex-matched controls. Our study partially confirmed the assumption. We found that competitive sport-involved adolescents demonstrated greater appearance and body areas satisfaction compared to controls. These results are in accordance with other studies that have demonstrated that sport-involved adolescents show greater body image satisfaction [15,16,17,22,46]. Some studies argued that adolescent sport participants might experience lower distress associated with physical appearance because they may come closer to an ideal of physical perfection than non-athletes [21]. Other studies in adolescents also show that greater physical fitness is associated with a more positive body image [47]. 

Further, we found no significant difference in DEABs in the groups of sport-involved adolescents and controls. Our results also coincide with those of other studies that demonstrated no differences in DEABs in sport-involved adolescents and controls [8,24,48,49]. However, our findings oppose the studies which concluded that the prevalence of eating disorders is higher in athletes compared to non–athletes [31,36]. This might be explained by the fact that the majority of studies in which associations between disordered eating and participation in a sport were observed in elite athletes. In our sample, we did not assess whether subjects competed at an elite level, and this may be considered to be a limitation of the study. Further studies should address this issue.

Next, we hypothesized that participants of the competitive sport would demonstrate greater body image and lower DEABs compared to leisure exercisers. Our study demonstrated no significant differences in body image of competitive athletes and leisure exercisers with exception for appearance evaluation. We found that competitive athletes evaluated their appearance more positively compared to leisure exercisers. Contrary to expectations, we found no significant differences in competitive athletes and leisure-time exercisers DEABs. As it is one of the first studies to compare DEABs in competitive athletes’ and leisure exercisers, the direct comparison of the results with other studies is limited. However, our results demonstrated that the type of involvement in sport (competitive vs leisure exercise) had no associations with DEABs in adolescents. Nevertheless, these findings should be tested in future studies. 

Further, we explored how sports classifications (weight-sensitive vs. less-weight-sensitive) are related to body image and the prevalence of DEABs. We expected that adolescents participating in weight- sensitive sports would demonstrate greater body image concerns and DEABs compared to participants of less-weight-sensitive sports. Analysis of competitive athletes demonstrated that there were no differences in body image concerns in weight-sensitive and less weight-sensitive groups. Further, no significant differences were found in DEABs of adolescents participating in weight-sensitive and less weight -sensitive competitive sports. Research in elite adolescent athletes also found no sport-specific differences in the prevalence of DEABs, and the researchers concluded that disturbances in eating behaviors were not limited to sports that emphasize leanness [32,37]. The same tendency was observed in some studies with adult athletes [50,51]. However, future studies should clearly divide competitive athletes into elite and non-elite groups and continue to explore this issue. 

However, the analysis of body image and DEABs in leisure time exercisers showed that adolescent females involved in weight-sensitive leisure sports reported greater body weight preoccupation and DEABs. It might be explained by the fact that a significant part of our leisure-time exercisers sample reported participating in leisure time fitness activities such as dancing, group fitness activities, weight training in gyms. This type of physical activity might be undertaken as a quest to improve appearance. The greater body image concerns and prevalence of DEABs in girls involved in weight-sensitive leisure sports might be explained by the fact that some girls with greater body image concerns might be motivated to exercise in leisure sports to improve their body image or to decrease body weight [52]. It was speculated that fitness centers’ environment (e.g., mirrors, images of fit female bodies, wearing of tight exercise clothing) might elevate body dissatisfaction in adolescent girls [53]. Our findings coincide with other studies that demonstrated that adolescent girls participating in aesthetic leisure sports reported significantly greater body image concerns [18]. As this is a cross-sectional study, it might also be that higher body image concerns and DEABs in girls involved in weight-sensitive leisure sports are an outcome of leisure exercising. Leisure-time sports, especially aesthetically focused, may accentuate one’s awareness of observers’ perspectives and the need to adhere to body ideals. Thus, participating in leisure aesthetically focused sports might be associated with adolescents placing less emphasis or value on physical competence or function and more emphasis on appearance. Studies demonstrated that adolescent girls who participated in aesthetic sports such as exercising in gyms, dancing, reported lower satisfaction in their bodies and its’ functionality [54], self- objectification, negative body esteem, and drive for thinness [53,55]. 

Further, our study demonstrated that boys participating in weight-sensitive leisure sports reported increased muscularity-motivated behaviour. This finding is quite paradoxical, yet it might be explained by the complex nature of adolescent boys’ body image. Adolescent boys’ participation in weight-sensitive leisure sports might be associated with the motivation to have a lean and muscular body. Future studies should continue to analyze the drive for muscularity in adolescents as studies have demonstrated that the drive for of muscularity is related to disordered eating [31]. 

To summarize, we found a gender–weight-sensitivity group interaction: the girls involved in leisure exercising in weight-sensitive sports demonstrated greater DEABs, and the boys involved in leisure exercising in weight-sensitive sports demonstrated a higher drive for muscularity than those involved in less-weight-sensitive leisure sports. Thus, as expected, gender moderated the associations between body image, DEABs, and characteristics of sports practice. These results reinforce the findings of other studies demonstrating a clear gender effect in associations between sport-related characteristics and DEABs [8,24,33]. However, no gender–weight-sensitivity group interactions were observed in competitive sports. Thus, the main novel finding of this study is that female gender in weight-sensitive leisure sports, but not competitive weight-sensitive sports, are associated with greater body image concerns and DEABs in adolescents. However, further studies should replicate these findings.

Our results indicate that being a participant in competitive sports over leisure exercising in weight- sensitive sports may be somewhat protective for body image concerns, which are related to normal physical changes during adolescence. In competitive sports, a body’s performance and functionality are more valued than a body’s appearance [56]. This might help athletes involved in competitive weight-sensitive sports to resist sociocultural pressure for the “ideal” body. Finally, in Lithuania, more qualified coaches are working in competitive sports than in leisure exercising (all coaches in competitive subsectors have higher education diplomas in sport studies). Therefore, it might be that they approach body weight managing and body image concerns- related issues more professionally than the fitness instructors or dance teachers who have less education in sport [57]. 

However, our findings should be replicated in other samples, and further studies should continuously focus on competitive weight-sensitive sports and explore body image and disordered eating issues. Participation in leisure sports is encouraged worldwide; however, our study adds to the knowledge that body image and eating behavior of adolescents participating in weight-sensitive leisure sports might be problematic. Therefore, special attention from public health specialists, as well as health and physical educators should be given to adolescents participating in dance, gym sports, and health and fitness activities in leisure time. Specialized training on topics of body image concerns and disordered eating prevention for coaches, fitness instructors, parents, athletes and leisure sport participants is highly recommended.

The first strength of the present study is the relatively large sample of adolescents of both genders. However, having even larger samples would enable researchers to assess the effect of age. The possibility to compare adolescents in groups of competitive and leisure sports is also a strong component of this research. It is also important that we explored not only DEABs, but the drive for muscularity, body image and health-compromising eating-related behaviours, including the use of diuretics, laxatives, and diet pills. These were recommended in other studies [8].

Several limitations must be accounted for when interpreting these results. First, the status of participation in sports was based on self-reporting. Further studies should study adolescents involved in organized sport settings (i.e., school sports) and compare the results with matched controls from general schools. Further, participation in sports at an elite level should also be considered in future studies to compare adolescents’ body image and prevalence of DEABs. DEABs in both genders might also be explored using more complex information (including BMI, drive for muscularity and drive for thinness, history of disordered eating, etc.) as reported in the other studies [4]. It would be reasonable to assess the body image and DEABs in the classification groups of weight-sensitive sports, i.e., in gravitational, weight class, and aesthetically judged sports [21] in larger samples in future studies. Finally, this study relied on a cross-sectional design; therefore, associations between variables are bidirectional. Future studies might benefit from a longitudinal design.

## 6. Conclusions

Our study resulted in several main findings. First, adolescents involved in sports demonstrated greater body image satisfaction and did not seem to present a greater risk for DEABs than those who were not involved in sports activities. Also, we observed no significant differences in DEABs comparing competitive athletes and leisure exercisers. Next, we found no differences or gender effects for body image concerns or DEABs in adolescents participating in weight-sensitive and less-weight-sensitive competitive sports. Finally, greater body image concerns and DEABs were more prevalent in girls participating in weight-sensitive leisure sports, and the motivation for muscularity was greater in boys participating in weight-sensitive leisure sports. Special attention from public health specialists, as well as health and physical educators should be given to adolescents participating in dance, gym sports, and health and fitness activities in leisure time. Special training on topics of body image concerns and disordered eating prevention for coaches, fitness instructors, parents, athletes and leisure sport participants is highly recommended.

## Figures and Tables

**Table 1 nutrients-11-03061-t001:** Descriptive statistics for the sample according to participation in sports (*n* = 732).

Characteristics		Participation in Sports			*χ* ^2^	*p*-Value
		None (*n* = 273)	Leisure (*n* = 239)	Competitive (*n* = 220)		
Gender (%)	Boys	27.4	35.1	37.5	23.4	<0.0001
	Girls	44.1	30.9	24.9		
Body weight (%)	Underweight	40.8	32.7	26.5	0.9	0.930
	Normal weight	36.7	32.7	30.6		
	Overweight/obesity	37.5	32.1	30.4		
Physical activity (%)	0 days/week	78.5	16.7	4.8	110.4	<0.0001
	1–2 days/week	57.1	31.6	11.3		
	3–4 days/week	25.2	37.4	37.4		
	5–7 days/week	27.3	29.9	42.8		
Exercise frequency (%)	0 days/week	-	69.7	30.3	22.8	<0.0001
	1–2 days/week	-	65.9	34.1		
	3–4 days/week	-	45.9	54.1		
	5–7 days/week	-	40.4	59.6		
Participation in weight sensitive sports (%)		-	65.3	60.9	0.9	0.333
Involvement in sports duration in years (mean ± SD)		-	2.9 ± 3.2	3.7 ± 3.3	20,641.0 *	0.002

*p* = significance level, SD = standard deviation; * Mann Whitney U test.

**Table 2 nutrients-11-03061-t002:** Disordered eating attitudes and behaviours, body image and self-esteem across three groups of sport participation, controlling by gender and body mass index (mean; 95% CI), (*n* = 732).

Characteristics	Participation in Sports			F	*η* ^2^	*p*-Value
	None (*n* = 273)	Leisure (*n* = 239)	Competitive (*n* = 220)			
Self-esteem	28.4 (27.9–29.0)	28.8 (28.3–29.4)	29.7 * (29.1–30.4)	5.1	0.014	0.006
Disordered eating	7.2 (6.2–8.4)	7.9 (6.7–9.1)	7.3 (6.1–8.5)	0.4	0.001	0.671
HCEBs	2.4 (1.9–2.8)	2.3 (1.8–2.8)	2.4 (1.9–2.9)	0.006	0.000	0.994
MBSRQ-AE	3.1 (3.1–3.2)	3.2 ^#^ (3.2–3.3)	3.4 ** (3.3–3.4)	13.0	0.035	<0.0001
MBSRQ-AO	3.3 (3.3–3.4)	3.5 ** (3.4–3.6)	3.5 ** (3.5–3.6)	12.0	0.033	<0.0001
MBSRQ-BAS	3.3 (3.3–3.4)	3.5 (3.4–3.5)	3.6 ** (3.5–3.7)	10.0	0.027	<0.0001
MBSRQ-OP	2.3 (2.2–2.4)	2.4 (2.3–2.5)	2.4 (2.3–2.5)	2.3	0.006	0.105
MBSRQ-SCW	3.0 (2.9–3.0)	3.0 (2.9–3.1)	2.9 (2.8–3.0)	1.4	0.004	0.253
DMS-B	0.6 (0.5–0.7)	0.8 ^#^ (0.7–0.9)	1.0 ** (0.8–1.1)	8.8	0.023	<0.001

* *p* < 0.01, ** *p* < 0.001 as compared to non -participating in sports; ^#^
*p* < 0.05 as compared to competitive sports group; *P* = significance level, HCEB = health compromising eating behaviors, F = Fisher’s statistics, *η^2^* = estimates of effect size, MBSRQ = the Multidimensional Body and Self Relations Questionnaire, AE = appearance evaluation, AO = appearance orientation, BAS = body areas satisfaction, OP = overweight preoccupation, SCW = self-classified weight; DMS-B = Muscle Development Behaviours.

**Table 3 nutrients-11-03061-t003:** Comparison of adolescents’ body image concerns and disordered eating (mean; 95% CI) across weight-sensitive and less-weight-sensitive leisure sports groups and genders* (*n* = 239).

Characteristics	Less-Weight-Sensitive Sports			Weight-Sensitive Sports			Weight-Sensitivity			Gender			Gender*Weight-Sensitivity		
	Boys *n* = 53	Girls *n* = 30	All *n* = 83	Boys *n* = 52	Girls *n* = 104	All *n* = 156	F	*η* ^2^	*P*	F	*η* ^2^	*P*	F	*η* ^2^	*P*
Self-esteem	29.9 (28.8–31.1)	29.7 (28.2–31.3)	29.8 (28.9–30.8)	28.3 (27.1–29.5)	28.2 (27.4–29.1)	28.3 (27.6–29.0)	6.5	0.027	0.012	0.07	0.000	0.799	0.005	0.000	0.943
Disordered eating	4.3 (1.9–6.8)	7.0 (3.8–10.2)	5.7 (3.7–7.7)	3.0 (0.5–5.4)	12.2 (10.5–14.0)	7.6 (6.1–9.1)	2.3	0.01	0.133	21.3	0.084	<0.0001	6.7	0.028	0.01
HCEB	0.9 (0.1–1.8)	1.8 (0.7–2.9)	1.4 (0.7–2.1)	0.9 (0.07–1.8)	3.7 (3.1–4.3)	2.3 (1.8–2.9)	4.8	0.02	0.029	16.7	0.067	<0.0001	5.0	0.02	0.026
MBSRQ-AE	3.2 (3.1–3.3)	3.3 (3.1–3.5)	3.2 (3.1–3.4)	3.1 (3.0–3.3)	3.3 (3.2–3.4)	3.2 (3.1–3.3)	0.3	0.001	0.586	2.9	0.013	0.087	0.2	0.001	0.626
MBSRQ-AO	3.2 (3.1–3.4)	3.5 (3.3–3.7)	3.4 (3.3–3.5)	3.3 (3.1–3.4)	3.8 (3.7–3.9)	3.5 (3.4–3.6)	4.0	0.017	0.046	24.2	0.096	<0.0001	3.0	0.013	0.085
MBSRQ-BAS	3.5 (3.4–3.7)	3.4 (3.2–3.6)	3.5 (3.3–3.6)	3.6 (3.4–3.8)	3.4 (3.3–3.5)	3.5 (3.4–3.6)	0.4	0.002	0.544	4.2	0.018	0.042	0.2	0.001	0.688
MBSRQ-OP	1.9 (1.7–2.1)	2.3 (2.1–2.6)	2.1 (1.9–2.3)	1.9 (1.6–2.1)	2.9 (2.7–3.0)	2.4 (2.2–2.5)	5.4	0.023	0.021	39.7	0.146	<0.0001	5.6	0.024	0.019
MBSRQ-SCW	2.8 (2.6–2.9)	3.1 (2.9–3.3)	2.9 (2.8–3.0)	2.7 (2.6–2.9)	3.2 (3.1–3.3)	3.0 (2.9–3.0)	0.2	0.001	0.680	24.3	0.095	<0.0001	0.7	0.003	0.405
DMS-B	1.0 (0.8–1.2)	0.4 (0.2–0.7)	0.7 (0.5–0.9)	1.5 (1.3–1.7)	0.4 (0.3–0.6)	1.0 (0.8–1.1)	4.3	0.017	0.039	43.9	0.151	<0.0001	4.5	0.018	0.036

* controlled for BMI, CI = confidence interval, *P* = significance level, F = Fisher’s statistics, *η*^2^ = estimates of effect size, HCEB = health-compromising eating behaviors, MBSRQ = the Multidimensional Body and Self Relations Questionnaire, AE = appearance evaluation, AO = appearance orientation, BAS = body areas satisfaction, OP = overweight preoccupation, SCW = self-classified weight; DMS-B = Muscle Development Behaviors.

**Table 4 nutrients-11-03061-t004:** Comparison of adolescents’ body image concerns and disordered eating (mean; 95% CI) across weight-sensitive and less-weight-sensitive competitive sports groups and genders* (*n* = 220).

Characteristics	Less-Weight-Sensitive Sports			Weight-Sensitive Sports			Weight Sensitivity			Gender			Gender*Weight Sensitivity		
	Boys *n* = 58	Girls *n* = 28	All *n* = 86	Boys *n* = 54	Girls *n* = 80	All *n* = 134	F	*η* ^2^	*P*	F	*η* ^2^	*P*	F	*η* ^2^	*P*
Self-esteem	29.0 (27.8–30.3)	29.0 (27.2–30.8)	29.0 (27.9–30.1)	29.3 (28.0–30.6)	30.8 (29.7–31.9)	30.1 (29.2–30.9)	2.2	0.01	0.136	1.0	0.005	0.315	1.1	0.005	0.291
Disordered eating	4.9 (2.7–7.1)	6.8 (3.5–10.1)	5.8 (3.9–7.8)	5.8 (3.5–8.2)	9.2 (7.2–11.1)	7.5 (6.0–9.0)	1.8	0.008	0.186	4.2	0.019	0.042	0.3	0.002	0.567
HCEB	1.4 (0.4–2.5)	2.9 (1.3–4.4)	2.2 (1.2–3.1)	1.0 (−0.2–2.1)	3.3 (2.3–4.3)	2.1 (1.4–2.9)	0.0	0.000	0.996	8.8	0.04	0.003	0.6	0.003	0.459
MBSRQ-AE	3.4 (3.2–3.5)	3.4 (3.2–3.5)	3.4 (3.3–3.6)	3.4 (3.3–3.5)	3.3 (3.2–3.4)	3.3 (3.3–3.4)	0.2	0.001	0.623	0.2	0.001	0.619	1.4	0.006	0.239
MBSRQ-AO	3.3 (3.2–3.5)	3.7 (3.5–3.9)	3.5 (3.4–3.6)	3.4 (3.2–3.5)	3.6 (3.5–3.8)	3.5 (3.4–3.6)	0.2	0.001	0.695	21.2	0.09	<0.0001	0.6	0.003	0.436
MBSRQ-BAS	3.7 (3.5–3.8)	3.7 (3.6–3.9)	3.6 (3.4–3.9)	3.6 (3.5–3.7)	3.7 (3.5–3.9)	3.5 (3.3–3.6)	0.5	0.002	0.467	2.9	0.013	0.092	0.5	0.002	0.493
MBSRQ-OP	1.9 (1.7–2.1)	2.6 (2.3–2.9)	2.2 (2.1–2.4)	1.9 (1.7–2.2)	2.8 (2.6–2.9)	2.3 (2.2–2.5)	0.7	0.004	0.383	34.4	0.139	<0.0001	0.6	0.003	0.441
MBSRQ-SCW	2.7 (2.6–2.9)	3.1 (2.9–3.3)	2.9 (2.8–3.0)	2.7 (2.6–2.9)	3.0 (2.9–3.2)	2.9 (2.8–3.0)	0.001	0.000	0.970	16.8	0.073	<0.0001	0.1	0.001	0.719
DMS-B	1.4 (1.2–1.7)	0.4 (−0.02–0.8)	0.9 (0.7–1.2)	1.7 (1.4–1.9)	0.6 (0.4–0.8)	1.1 (1.0–1.3)	2.0	0.009	0.156	48.1	0.185	<0.0001	0.0	0.000	0.986

* controlled for BMI, CI = confidence interval, *P* = significance level, F = Fisher’s statistics, *η*^2^ = estimates of effect size, HCEB = health-compromising eating behaviors, MBSRQ = the Multidimensional Body and Self Relations Questionnaire, AE = appearance evaluation, AO = appearance orientation, BAS = body areas satisfaction, OP = overweight preoccupation, SCW = self-classified weight, DMS-B = Muscle Development Behaviors.

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
