# Peer review of "Body Image and Disturbed Eating Attitudes and Behaviors in Sport-Involved Adolescents: The Role of Gender and Sport Characteristics"

_nutrients, 2019, doi:10.3390/nu11123061_

Round 1

Reviewer 1 Report

The presented study is interesting and well presented. Although, some issues must be addressed:

In some sentences, the expressions used or the structure of the phrase difficult the reading and understanding the meaning. This issue has been spotted on the introduction. Example: lines 42-44.  In the methods section, the authors referred to thee groups regarding sport (none, leisure and competitive sports), though the type of exercise of each group has not been describe. Moreover, the number of adolescents per group is not described.  In the methods section, the authors describe that they used several parametric and non-parametric tests, although the exact test is not described. In addition, the test of normalization (Shapiro-Wilks or Kolmogorov-Smirnov) has not been included or described depending on each group and therefore the test used. The results are focused on the 4 tables, but the description and presentation of the data in a systematic and detailed way has been reduced. This sections seems to be missing the highlights and comments on the themes that emerge from the analysis.  The conclusion section seems to be shorter than expected from the study. This conclusion could be resulted from the reduction in the results section. 

Author Response

Dear Reviewer,

Thank you for your comments and remarks. We updated and improved our paper in accordance to them. All the corrections and updates in the text are highlighted in blue color. Please find summary table with our responses attached.

Reviewer 2 Report

Interesting data collected, and the current study has potential to add to the literature, though not in its current form. The following edits are suggested:

Introduction:

1) Stylistic, and grammar details should be corrected. For example, E.g. needs commas after; Typos (e.g., Kong & Harris).

2) Lines 42-44 speak about disordered eating in adolescents in general; lines 45-48 argue that adolescent athletes don’t eat appropriately and have low body fat.  Taken together, they don’t really support the argument that ends the paragraph (i.e., that studying adolescent athletes and DEABs is necessary). I think this argument would be more strongly supported by citing studies of adolescent athletes who display eating pathology – not just poor nutrient intake.

3) Re: the paragraph about body image, I might steer clear of citing studies of adult athletes and instead, stay focused on body image and adolescents as you do in the following paragraph. I would also suggest cutting lines 83-86, for the same reason.

4) The paragraph on adolescent body image in athletes – could it be structured to be a little more organized for the reader?  While the results of the extant literature are not convergent, the way this is presented needs a little more structure.

5) You acknowledge the novelty of this study in Eastern Europe. If it is not otherwise novel, it would be good to think about and include some ways in which you believe that the cultural differences specific to this population would be unique, and necessary to study. I am not convinced that the current study adds to the literature in a meaningful way, so thinking about how to present it would be important.

6) The structure of the introduction overall is difficult to follow.  I think it would benefit from some shortening, and removal of unnecessary information.  Also, it is unclear as I am reading through it, exactly what analyses will be tested.  The flow of the text should lead me, as the reader, to already know what you are going to test, before I get there – and that is not clear here.

7) Do you have hypotheses? These should be included more definitively at the end of the introduction. You present your questions, but not what you expected to find. These will also help to guide what results you choose to present.

Methods:

8) Questionnaires that are incomplete could be treated as missing data – why did the authors choose not to use these questionnaires, instead?

9) Was there an empirical reason for assuming that ½ year of sport participation was a requirement?

10) Lines 135-136: can you explain the standard by which people fit into the specific weight classes?

11) I might have missed this, but it is very important that somewhere in the manuscript (or an appendix), the authors list all of the sports, the sample size for each, and how the determination of weight sensitive was made, specific to each sport. The classification was cited, but as a field, it is critical that we know what we are classifying.

Results:

12) Table 1 is a bit confusing as to what variables drive the significance. For example, for the first category (gender) the p value is indicated for both – which one, or did both, indicate significance?

13) In Table 2 -4 – the authors list disordered eating – is this the DEABs that they report as non- significant in text? It would help if the nomenclature is consistent across reporting. Also, the way the tables are organized is difficult to read. I suggest that the authors consider making more columns to add the F, eta-squared and p values (rather than making them all rows).

14) The results feel a bit like the authors tested a little bit of everything.  I understand that it is hard to decide what the most important questions are to ask, but I would appreciate a more focused analytic plan where the analyses are more focused, and based on hypotheses.  Perhaps consider presenting less, but presenting it in a clearer and more focused manner.

Discussion:

15) The paragraph that begins on line 299 gives information that might be better positioned in the introduction. Presenting new information in the discussion should be more focused on only explaining results (and succinctly).

16) The terms “weight sensitive,” “weight-class dependent,” “leanness-dependent,” “aesthetic” and “anti-gravitational” are only a few of the many terms used in the sporting literature to organize sports. The authors should provide more focused discussion on how their findings contribute to this panoply of terms – rather than adding to the conflicting results.

17) The discussion is quite long, but it could be more concise and focused. It would be helpful to also acknowledge how this study helps to clarify some of the conflicting results across the broader literature.

Author Response

Dear Reviewer,

Thank you for your valuable comments and remarks. We put our maximum efforts to update and improve our paper in accordance to them. All the corrections and updates in the text are highlighted in blue color. Please find summary table with our responses and explanations attached. Thank you once again for your contribution and help to improve out paper.

Round 2

Reviewer 1 Report

The authors had followed the suggestions made in the previous step. Nevertheless, some issues continue. 
1. First of all some expression could be modified throw in the results section to facilitate the reading.

2. In the methodology section, the sample given is 736, though when the different samples (220,239,273) are summed up the result is 732. Therefore, there are four missing data. Were these 4 missing not included in the statistical analysis?

Author Response

Dear Reviewer,

Thank you for your remarks and help to improve our paper.

We added information about 4 missing cases in the methods section. These study participants did not indicate the exact sport they participate in, thus we couldn't attribute them to weight sensitive or less weight sensitive group.

In addition, we went thorough the expressions in the results section concerning disturbed eating attitudes and behaviors and corrected them.

Sincerely,

The corresponding author

Reviewer 2 Report

Thank you for attending to typos and details.  There are still some grammar issues (lack of period after e.g. in blue fonted text, and misuse of apostrophes). I might also suggest that you consult the APA standards for reporting statistics (e.g., no leading zeros for p values).

Thank you for the explanation regarding why 6 months of sport participation was required.  I believe this citation and explanation should be included in the manuscript as well.

Page 5 – Man and Bonferroni are misspelled.  What was the Bonferroni adjustment? This should be specified in the text.

Table 1: Thank you for explaining what the tests were in your response and in text, but the table is still less clear.  In particular, the p value for the most far-right column is not adequately explained as to which groups are significantly different from one another. 

Tables 3 and 4: I appreciate the effort it takes to reconfigure tables, but I respectfully disagree that “adding additional columns makes statistics confusing.”  I suggest a landscape format, and the addition of at least 3 new columns.

Author Response

Dear Reviewer,

Thank you very much for your remarks.

We corrected grammar issues, included suggested reference and reworked Tables 3 and 4 following your recommendations. Moreover, we added chi square test to the Table 1, which clarifies that distributions of different study variables were compared across 3 participation in sports groups. We accept your remark that comparing proportions in pairs complicates table even more and are excessive, so we removed them.

According to the Nutrients journal requirements, statistics must be presented in a style common to biomedical journals, not APA.